# Key Role of Astrocytes in Postnatal Brain and Retinal Angiogenesis

**DOI:** 10.3390/ijms23052646

**Published:** 2022-02-28

**Authors:** Mariela Puebla, Pablo J. Tapia, Hilda Espinoza

**Affiliations:** 1Centro de Fisiología Celular e Integrativa, Facultad de Medicina-Clínica Alemana, Universidad del Desarrollo, Av. Plaza 680, Las Condes, Santiago 7550000, Chile; mpueblac@gmail.com; 2Centro de Biología Celular y Biomedicina (CEBICEM), Facultad de Medicina y Ciencia, Universidad San Sebastián, Av. Lota 2465, Providencia, Santiago 7500000, Chile; pablo.tapia.o@gmail.com; 3Facultad de Medicina Veterinaria y Agronomía, Universidad de las Américas, Av. República 71, Santiago 8320000, Chile; 4Facultad de Ciencias de la Salud, Universidad del Alba, Av. Ejército Libertador 171, Santiago 8320000, Chile

**Keywords:** brain angiogenesis, retinal angiogenesis, astrocytes, endothelial cells, extracellular matrix

## Abstract

Angiogenesis is a key process in various physiological and pathological conditions in the nervous system and in the retina during postnatal life. Although an increasing number of studies have addressed the role of endothelial cells in this event, the astrocytes contribution in angiogenesis has received less attention. This review is focused on the role of astrocytes as a scaffold and in the stabilization of the new blood vessels, through different molecules release, which can modulate the angiogenesis process in the brain and in the retina. Further, differences in the astrocytes phenotype are addressed in glioblastoma, one of the most devastating types of brain cancer, in order to provide potential targets involved in the cross signaling between endothelial cells, astrocytes and glioma cells, that mediate tumor progression and pathological angiogenesis. Given the relevance of astrocytes in angiogenesis in physiological and pathological conditions, future studies are required to better understand the interrelation between endothelial and astrocyte signaling pathways during this process.

## 1. Introduction

Blood vessel growth during early postnatal brain development requires an increase in progenitor cells and their differentiation into astrocytes in the central nervous system [1]. Strong coordination between the astrocyte differentiation and the vessel growth during this stage of brain development has been shown and, in fact, angiogenesis (process where new blood vessels are formed from pre-existing vessels) and astrogenesis occur almost at the same time [2]. In vitro studies have confirmed the relevance of astrocyte differentiation for the development of brain vasculature and the importance of establishing co-culture models of astrocytes and endothelial cells (ECs) to study interactions between these cells during angiogenesis [3].

Specific genetic inhibition of astrogliogenesis in the early postnatal mouse cortex resulted in an important delay in blood vessel growth and branching, which is consistent with the critical role of astrocytes in early postnatal brain development [4]. Moreover, astrogliosis, a complex process with positive and negative effects associated with the expression of many genes and morphological changes in astrocytes when an insult in the central nervous system occurs, has been shown to be necessary to restore blood vessel morphology by increasing proangiogenic signaling mediated by astrocytes, but negative effects have also been described [1,4]. Several studies have been focused on determining the role of astrocytes in the angiogenic process from a mechanistic point of view. Many of these studies have used the murine retina, an excellent model to study the signaling related to angiogenesis in the central nervous system, particularly the signals released from astrocytes during this process, due its easy anatomical accessibility. Angiogenesis seems to depend on the communication between different cells, such as neurons, glia, ECs, pericytes, and even immune cells in the brain and in the retinal angiogenesis [5,6,7,8]. These studies show the relevance of the astrocytes in the angiogenesis process during the brain and in retina development angiogenesis, which seems to have similarities regarding the complex mechanisms involved, and where the astrocytes play a key role.

In this review, we describe the information available about the role of astrocytes in the angiogenic process, during postnatal brain and retinal development, with emphasis on their role as angiogenic guides, the release of pro- or antiangiogenic molecules from astrocytes and the feedback mechanism between astrocytes and blood vessels.

## 2. Astrocytes as Templates for Angiogenesis

Astrocytes are the most abundant cell type in the brain; they appear at the later stages of brain development, with maximum astrogenesis in the first postnatal days [9,10]. According to this, astrocytes have an important role in angiogenesis during postnatal development and participate in the consolidation of the primary vasculature in the brain, acting as a template over which ECs migrate to form new blood vessels [4,9]. In addition to the relevance of the astrocytes in the brain angiogenesis process, in numerous studies using the mouse retinal model, researchers have shown that astrocyte scaffolding functions as a template for vascular network development in the retina [11,12,13]. In fact, the disruption of the angiogenic astrocyte template induces alterations in the retinal angiogenic process [14]. Similar to what was observed in the brain, Phng et al. reported that during a three-week period after birth, ECs migrate over a preexisting astrocytic template and form new blood vessels in the mouse retina [15].

Various cell adhesion molecules as integrins, cadherins and laminin protein complexes from astrocytes, have been associated with their role as scaffolding for angiogenesis, and their role will be described below.

### 2.1. Cadherins: A Possible Role as an Angiogenic Cue

Cadherins are a calcium-dependent family of transmembrane proteins with an important function in cell to cell adhesion, awarding stability and mechanical resistance, and playing an important role in the morphogenesis and homeostasis of tissue [16]. There have been described different types of classical cadherins, named according their first reported localization despite that they are not exclusively expressed in that tissue: N-cadherin (neural), P-cadherin (placenta), T-cadherin (heart), VE-cadherin (vascular epithelial), and R-cadherin (retinal) [17], but we will focus in some which are expressed in astrocytes and would be related with the angiogenesis process in the brain and retina.

Perinatal astrocytes and oligodendrocytes express N-cadherin on their surfaces, and the blockade of N-cadherin increases the migration of oligodendrocytes on an astrocyte cell surface, suggesting that the cell interaction mediated by this cadherin has a negative effect on cell migration [18]. Moreover, it has been described that endothelial signaling through N-cadherin reduces the polarized phenotype of migrating smooth muscle cells, suggesting a role in blood vessel stabilization [19].

In addition, R-cadherin has shown an important function in retinal morphogenesis, and is expressed in glial cells and in early optic nerve glia in the rat optic nerve, which is an immature kind of astrocyte [20]. It has been demonstrated that N-cadherin axons elongate on using glial R-cadherins as substrate [21]. During the postnatal stage, R-cadherin expression is higher between postnatal days 0 to 4 (P0 to P4) than postnatal day 8 (P8) in the retinal astrocytic process surrounding new vessels, and coincides spatially and temporarily with the formation of the vascular network. This suggests that small R-cadherin clusters represent sites of endothelial filopodial extension. The blockade of R-cadherin prevented the normal extensive collateralization in the superficial vessel network and had negative effects in the deep vessel network [22], and this could indicate that R-cadherin from the astrocytes template serves as a cue for ECs migration during angiogenesis.

An atypical cadherin, FAT1 cadherin, is expressed in the nervous system with an important function in neural differentiation [23], and also in the regulation of cell polarity and migration [24]. FAT1, and one of its signaling pathways, Hippo, have been associated with the early stages of neurogenesis, but their regulation is not completely understood [23]. In addition, Fat1 is also expressed in retinal astrocytes during postnatal life [25]. Consistent with that, the elimination of Fat1 in postnatal retinas impaired the association of ECs and astrocytes, proliferation and migration of astrocytes progenitors, and their maturation into immature astrocytes, and delays postnatal angiogenesis [25]. This is in line with a previous report from Caruso et al. (2013), where *Fat1* mutant mice displayed retinal microvascular abnormalities as microaneurysms [26]. However, if this cadherin is involved in angiogenesis in the brain, it has not been addressed.

In summary, cadherins such as R-cadherin and the atypical Fat1 cadherin promote angiogenesis in the retina, and N-cadherin seems to be relevant for blood vessel stabilization. Future studies are needed to characterize the signaling pathways between ECs and astrocytes mediated by cadherins during angiogenesis.

### 2.2. Fibronectin and Integrins in Scaffold Formation

Fibronectin is an important component of the extracellular matrix which permits cell adhesion through the transmembrane receptors named integrins, participating in the cell proliferation process [27].

Astrocytes are the major source of fibronectin during retinal angiogenesis. A study conducted by Stenzel et al. supports the idea that fibronectin secreted by astrocytes provides a scaffold for guiding directional angiogenesis into the avascular region of the retina, and the key role of this mechanism during the retinal angiogenic process [28]. In fact, the specific deletion of fibronectin from astrocytes resulted in not only a reduction in ECs migration, but also a reduced signaling through phosphatidylinositol 3-kinase/protein kinase B or PKB (PI3K/Akt signaling), and the vascular endothelial growth factor receptor-2 (VEGFR-2) expression [28]. PI3K is activated by growth factors such as endothelial cell growth factor (VEGF), which is a recognized angiogenic molecule in physiological and physiopathological conditions [29]. The binding of VEGF to VEGFR2 in ECs, activates the downstream kinase Akt to promote angiogenesis by activating mammalian target of rapamycin (mTOR). This pathway may have a key role in the proliferation, adhesion, and migration of ECs during the angiogenic process [30,31]. In accordance with the relevant role of fibronectin in angiogenesis, the inhibition of fibronectin production from astrocytes decreased the activation of VEGFR2 and consequently PI3K/Akt signaling, which had negative effects on filopodia adhesion to the extracellular matrix and tip cell migration; as result, the radial expansion of the vascular plexus was reduced [28].

During normal retinal development, the high cell proliferation rate produces physiological hypoxia that activates hypoxia inducible factors (HIF), which induce angiogenesis to ensure the appropriate oxygen levels and nutrients in the developing tissues [32,33]. Astrocytes may act as oxygen sensors, and hypoxia during retinal development could induce astrocyte proliferation to establish an adequate template for retinal angiogenesis [14]. In this context, the orphan nuclear receptor tailless (TLX) is one of the mediators downstream of HIF signaling produced by astrocytes that has an important role during retinal development [34]. TLX (NR2E1), a nuclear receptor that acts as a transcription factor, is predominantly expressed in the central nervous system, and has a critical role in the embryonic and adult neurogenic process and neural development, especially in the visual system [35,36,37]. Studies from Uemura et al. showed that TLX expression increases in proangiogenic astrocytes and has a crucial role in the formation of the extracellular scaffold formed by these astrocytes through the upregulation of the expression and extracellular deposition of fibronectin, which provides the optimal conditions for survival, adhesion, and migration of ECs [34]. In fact, genetic ablation of Tlx abrogated normal retinal vascular development, as evidenced by deficiencies in the deposition of fibronectin, which produced a disorganized architecture of the astrocyte network and abnormal angiogenesis during the postnatal stage in this model. Additionally to the fibronectin deposition surrounded astrocytes networks dependent of Tlx signaling, an increased α5 and β1 integrin subunits expression in ECs was observed, which is consistent a role of fibronectin/integrin signaling in the adhesion of migrating ECs to the astrocytic scaffold during the development of the retinal vascular system [35]. Moreover, the binding of angiopoietin-1 (Ang-1), a secreted glycoprotein that is part of the angiopoietin family of growth factors, has an important role in vascular development, remodeling, and stabilization, through its binding to tyrosine kinase receptors in ECs [38]. It is known that Ang-1 is secreted by astrocytes in hypoxic conditions (see below), and it activates the αvβ5-focal adhesion kinase (FAK)–AKT signaling pathway in retinal astrocytes and stimulates fibronectin secretion [39]. These antecedents suggest that hypoxia via Tlx pathway or/and Ang-1 pathway promote the fibronectin secretion from astrocytes that later interacts with integrins in ECs promoting retinal angiogenesis.

Furthermore, to the role of fibronectin in the retinal angiogenesis, it has been reported that fibronectin is essential for angiogenesis in the brain, promoting the survival and also the proliferation of capillaries ECs by the activation of the mitogen-activated protein kinase (MAPK) signaling pathway through integrin receptors α5β1 and αvβ3 [40]. These observations are consistent with the overexpression of fibronectin and its receptors α5β1 and αvβ3 in angiogenic vessels, as response to an ischemic event in the brain in murine models [41,42]. This suggests that similar to what occurs during retinal angiogenesis, the interaction between integrins and fibronectin during hypoxia stimulates brain angiogenesis.

Consistent with the role of integrins in angiogenesis, the correct retinal vascular plexus formation in the mouse retina has been shown to be sensitive to the ablation of the αvβ8 integrin expressed in retinal astrocytes, evidenced by the impairment of the development of the vascular plexus and a significantly lower number and length of filopodia in tip ECs. This phenomenon was related to the lack of activation of the transforming growth factor β (TGFβ) signaling in ECs and highlights the importance of the αvβ8 integrin from astrocytes in the control of retinal angiogenesis [12]. In contrast with these findings, in cocultures of astrocytes and brain ECs, the astrocytic αvβ8 activation was dependent of TGFβ, and a possible signaling through their receptors induced the expression of two known antiangiogenic molecules in ECs, the plasminogen inhibitor-1 and thrombospondin-1, and inhibited endothelial migration [43]. This suggests that in the brain, the αvβ8 integrin-TGFβ pathway has a role in blood vessel maturation and stabilization.

### 2.3. Laminins as a Template for Angiogenesis

According to the relationship of the extracellular matrix to astrocyte signaling during angiogenesis in retinal models, the loss of other component of the extracellular matrix, as laminin, and integrin, as was previously described, also had negative effects on the retinal angiogenic process [44,45].

Laminins are a glycoprotein family that are present in the extracellular matrix and are part of the basement membrane. These proteins can bind to their transmembrane receptors, the integrin family, expressed in many types of cells and mediate many processes in adhesion-mediated events in vertebrates [46]. Deletion of laminins β2 and γ3 in mouse models was shown to reduce astrocyte migration, probably through the reduction of astrocyte expression of β1 integrin, indicating the importance of laminin-integrin β1 interactions for laminin-directed astrocyte migration. Furthermore, deletion of laminins reduced the interaction between astrocytes and ECs and affected normal retinal blood vessels [44]. Moreover, it has been described that assembly of laminin network with proteoglycans are needed for the astrocytic migration and angiogenesis [47]. Further, laminin released by astrocytes seems to recruit and activate the microglia, and changes in the microglia contribute to angiogenesis in retinal models [45]. These results highlight the relevance of laminin-β1 integrin signaling and laminin-proteoglycan interaction to promote astrocyte migration and form the template for angiogenesis [44]. Future studies to elucidate the role of laminin in this process in depth greater are required.

The representation of the principal proteins (as fibronectin, cadherins, integrins, laminin, and others) and its receptors previously described, associated with astrocytes as templates for angiogenesis, is summarized in Figure 1.

In addition to the participation of the extracellular matrix proteins and cell adhesion molecules described above, various studies have focused on the mediators released from the astrocytes that contribute to the angiogenic process during brain development and in the retina. The next section describes the principal molecules produced by astrocytes and their roles in angiogenesis.

## 3. Pro- or Antiangiogenic Factors Released by Astrocytes

As has been previously described, vascular plexus formation depends on and follows the astrocytic plexus, but this is not the only mechanism through which astrocytes participate in the angiogenic process. Astrocytes release numerous signaling molecules that participate in the formation of the blood brain barrier [48], and the release of pro- or antiangiogenic factors from astrocytes was suggested to regulate angiogenesis [9,49].

### 3.1. Release of VEGF from Aastrocytes as a Proangiogenic Stimulus

Astrocyte synthesis and release of VEGF may be essential for the formation and growth of the vascular bed in the brain. In fact, VEGF can act as a neuroprotector mediator and stimulates the angiogenesis after an ischemic brain insult. This is in line with studies where the specific downregulation of VEGF expression in astrocytes has been shown to induce alteration in the development of the scaffold vasculature in the rostral migratory stream and create defects in neuroblast migration, demonstrating that VEGF secretion from astrocytes is essential during early postnatal brain development [50]. Consistent with the role of VEGF secreted by astrocytes in angiogenesis, it has been demonstrated that astrocytes in hypoxic conditions secrete higher levels of VEGF than in normoxic conditions, and in co-cultures of astrocytes and brain microvascular ECs decreased the apoptosis of ECs and also stabilized endothelial tubular structures under hypoxic conditions in response to VEGF [51]. This could suggest that the VEGF secretion from astrocytes in avascular areas of the brain, where the concentration of oxygen is lower, could be a key factor for the formation and stabilization of new blood vessels.

Similar findings have been observed in the assessment of angiogenesis in the retina. During astrocyte migration from the optic disk to the retina, the expression of VEGF in peripheral astrocytes is higher than that in astrocytes closer to the disk and precedes angiogenesis. Furthermore, physiological levels of hypoxia induce the secretion of VEGF from astrocytes and stimulate the development of retinal vessels, which is reduced by hyperoxia [52]. Consistent with this finding, it was shown that the VEGF gradient generated by astrocytes is sensed by long filopodia from ECs that form vascular sprouts and mediate directional endothelial migration to form new vessels, a response mediated by VEGFR2 [53].

Interestingly, two different studies have indicated that VEGF secretion from astrocytes is relevant in pathological conditions, but has a minor impact on retinal angiogenesis in postnatal life. Scott et al. showed that VEGF deletion in retinal astrocytes reduces the area covered by the vascular network in retinas from postnatal day 4 (P4) mice but not from postnatal day (P10) mice, suggesting a delay but not an impairment of vascular development. However, in mice exposed to hyperoxia, deletion of VEGF in astrocytes dramatically increased obliterated capillaries from retinas, an effect that was independent of HIF-1a [54]. Additionally, Weidemann et al. showed that an increase in VEGF secretion from astrocytes mediated by HIF-2a is essential for hypoxia-induced neovascularization but not for normal development of retinal vasculature, and deletion of the von Hippel-Lindau gene (a negative regulator of HIFa) increased VEGF expression in astrocytes to the same extent as that observed in hypoxic retinas [55]. In the latter study, the lack of effect observed in normal angiogenesis when VEGF expression was eliminated in astrocytes was attributed to a possible compensatory mechanism from other retinal cells (pericytes or ganglion cells), that could secrete VEGF to ensure normal angiogenesis [55]. Despite this explanation, a recent study conducted by Rattner et al. using the line based in the glial fibrillary acidic protein (GFAP, an intermediate filament protein, expressed in many cell types of the central nervous system, especially in astrocytes), the (*Gfap*)-Cre/*Vegf*^fl/fl^, the same strategy used by Scott et al. and Weidemann et al., eliminated the VEGF expression in astrocytes only in 78% of recombined astrocytes in the central vascularized retina, and only 40% of recombined astrocytes in the avascular peripheral retina. This suggests that the lack of effect observed in angiogenesis was related to a sufficient number of astrocytes that could express VEGF to lead to normal vascular development. In the same study and with another *Gfap*-Cre line that recombines more than 99% of retinal astrocytes, the elimination of VEGF production from astrocytes inhibited radial migration of retinal ECs and angiogenesis despite the presence of a fully formed astrocyte plexus [56]. This finding is consistent with the severe defects observed in radial migration of retinal ECs when HIF-2a was deleted in retinal astrocytes using the same *Gfap*-Cre line [57].

In summary, there is strong evidence showing that VEGF secreted by astrocytes serves as a guide for the angiogenic process during retina and brain vascular development. In addition, the oxygen levels are a key determinant of astrocytic behavior during angiogenesis and their VEGF secretion, representing a reciprocal fine-tuning between ECs and astrocytes during this process (see below).

### 3.2. Role of FGF-2 from Astrocytes during Angiogenesis

Fibroblast growth factor 2 (FGF-2) is part of the extended family of growth factors that participate in the regulation of key events such as proliferation, migration and differentiation during embryonic development and tissue repair during adult stages [58]. This factor has an important role in neurodevelopment, and alterations in its signaling pathway have been associated with neuronal loss and glial dysfunction during the postnatal stage of brain development, which has been related to psychiatric disorders [59].

In the central nervous system, FGF2 is the main isoform of FGF expressed in neuronal and glial cells, its receptor FGFR2 and FGFR3 are expressed in glial cells, while FGFR1 is predominantly expressed in neurons and endothelial [60,61]. The signaling through FGF receptors has been implicated in astrocytic differentiation, has described in studies of Irmady et al., where a reduction in GFAP levels in astrocytes in FGF2 null adult mice was observed, from pons and medulla oblongata, which coincides with a reduction in levels of FGFR2 but not of FGFR3 receptors in astrocytes, and a decreased STAT-binding site (signal transducer and activator of transcription) of GFAP promoter in FGF2 null mice. That suggests that FGF2 through FGFR2 signaling stimulates the differentiation of astrocytes in the hindbrain. This is consistent with a recent study from Savchenko et al., where FGF2 treatment of astrocytes upregulates genes involved in astrocytic growth and maturation [62].

In addition to the reduction of GFAP levels, in cortical astrocytes from FGF-2^−/−^/FGF-5^−/−^ double mutant mice, a diminish in levels of tight junction proteins Occludin and ZO-1 (Zona Occludens-1) in blood vessels and a reduction of levels of intermediate filaments in perivascular astrocytes endfeet was observed. That phenotype was rescued by FGF-2 exogenous administration, indicating an impact of this fibroblast growth factor in blood-brain barrier integrity [60]. On the other hand, an enhancement of GFAP levels was observed in astrocytes from spinal cord, hindbrain and cerebellum of mice lackingFGFR3, suggesting that FGFR3 signaling negatively regulates astrocytes differentiation [63].

The participation of FGF2 in angiogenesis through the FGFR1 signaling in endothelial cells, and stimulating endothelial proliferation, migration and tubular formation has been extensively studied in vivo and in vitro conditions [64,65,66,67,68]. In cortical organotypic cultures of mice it has been demonstrated that FGF2 treatment increases the number of new blood vessels [69], and in brain endothelial cells in culture, FGF-2 stimulates their proliferation and differentiation [70]. However, if FGF2 released from astrocytes participates in angiogenesis, it has not been addressed. It is interesting to note that during postnatal brain development, FGF-2 appeared in astrocytes at P4 in intracellular localization, and in the nuclear area during postnatal days 10 or 20 (P10 or P20), indicating that astrocytes produce FGF2 during postnatal life [71], and, more importantly, it has been described that astrocytes isolated from brain cortices have the ability to release vesicles that contain FGF-2 and VEGF [72]. Although the signals that induce the release of these vesicles have not been clarified, it is possible that they share equivalent signals related to VEGF release in astrocytes. An interesting study from Ganat et al. shows an augmentation of FGF2 in astrocytes from adult brain during chronic hypoxia exposure that correlates with a reduction in GFAP, but maintained expression of vimentin in these cells, suggesting that hypoxia transforms astrocytes into more immature glial cells [73]. In line with what was described above, we hypothesize that: (1) hypoxia could mediate an increase of production and secretion of FGF2 from astrocytes in the same manner that stimulate VEGF secretion [51,52], and FGF-2 acts as a pro-angiogenic stimulus for endothelial cells; and (2) the transition of astrocytes to immature phenotype in hypoxic areas could be associated with a similar modulation of angiogenesis that exerts radial glial cells during angiogenesis in developing central nervous system; but if this occurs during postnatal angiogenesis it remains to be elucidated.

Regarding angiogenesis in retina, in ex vivo models of choroidal neovascularization, FGF-2 stimulates the vascular sprouting via STAT3 activation [74], and in cultures of chorocapillary ECs, the stimulation with FGF2 induces cell proliferation through PI3K, MEK1 and ERK1/2 activation [75]. Furthermore, a loss of astrocytes and microglia was observed during hypoxic insults in retina, but a rescue of astrocytes was observed through intravital injection, exposition to astrocyte conditioned media that contain VEGF and FGF-2, or direct injection of FGF-2 which normalized the revascularization of vaso-obliterated retina [76]. This suggests a role of astrocytes and FGF derived from astrocytes in the development of normal vascular plexus after ischemic insult. Altogether, these studies suggest that VEGF and FGF-2 secreted together could mediate angiogenesis in the retina and brain following the activation of their growth factor receptor in ECs.

### 3.3. Participation of Angiopoietins Secreted by Astrocytes in Angiogenesis

Angiopoietins are members of a growth factor family with a critical role in the angiogenic process under physiological and even pathophysiological conditions [77]. The role of Angs and tyrosine kinase with immunoglobulin-like and EGF-like domain receptor type 2 (Tie-2) in cerebral angiogenesis has been previously addressed [78,79,80].

It has been suggested that Angs could participate in vascular remodeling during brain development. In particular, Ang-1 participates in vessel maturation through the modulation of ECs migration, adhesion, and survival. Additionally, another angiopoietin, angiopoietin-2 (Ang-2), may participate in the neovascularization process associated with the VEGF pathway [81]. Both Ang-1 and Ang-2 are ligands of the tyrosine kinase receptor Tie-2, and this receptor has been shown to be essential for vascular maturation during development [81]. In fact, the specific genetic ablation of Ang-1 and its receptor Tie-2 was shown to be lethal in the early embryonic stages in mouse models [77].

Furthermore, consistent with the pattern of cerebral cortical vascularization from deeper regions to outer regions, the expression of VEGF and Ang-1 in astrocytes of the cerebellar cortex follows an inside-out pattern, suggesting a role of Ang-1 in this process [82]. However, Ang-2 and tyrosine kinase with immunoglobulin-like and EGF-like domains receptor type 1 (Tie-1) were expressed in ECs of meningeal and cerebellar vessels, and an upregulation of endothelial Ang-2 mRNA expression was observed after middle cerebral artery occlusion, that coincides with endothelial proliferation in the peri-infarct area, whereas Ang-1 mRNA was constitutively expressed in astrocytes and neurons [82,83]. Further, in a model of hypoxic-ischemic brain injury, the transplantation of mesenchymal stem cells which activated astrocytes induced the secretion of proangiogenic factors, including Ang-1 and Ang-2, and stimulated endogenous angiogenesis in the neostriatum, with a significant number of astrocytes surrounding new blood vessels [84]. In accordance with this result, an increase in VEGF and Ang-2 expression in astrocytes dependent on the morphogen Sonic Hedgehog (Shh)/nuclear receptor subfamily 2 group F member 2 (NR2F2) pathway was observed in response to oxygen-glucose deprivation, a condition that mimics ischemic insult [85], but in the same model, Shh secreted by activated astrocytes induced proliferation, migration, and tubular formation from ECs, possibly through the activation of RhoA/Rho associated protein kinase (RhoA/ROCK) pathway in ECs, and not due to other proangiogenic molecules secretion [86]. The Shh signaling between astrocytes and endothelial cells has been described in the context of brain injury and repair, specifically in the maintenance of the blood brain barrier and between astrocytes and neurons modulating synaptic function and neuronal activity [87,88]. However, if this mechanism is implicated in the participation of Angs released from astrocytes in physiological angiogenesis, it has not been completely elucidated. Moreover, it was observed in cocultures with bone marrow stromal cells (MSCs), an increased expression of VEGF, Ang-1 and Tie-2 receptors in astrocytes, and conditioned media of MSC-astrocyte co-culture induced endothelial capillary tubular formation and inhibition of Ang1 or knockdown of Tie-2 in ECs reduced angiogenesis [89], this suggest that Ang-1 secretion by astrocytes could stimulate angiogenesis through endothelial Tie-2 activation.

In addition, the loss of Ang-1 and its downstream Tie2 signaling in the vascular region and integrin αvβ5 signaling in the astrocytes is a conduit to altered retinal vasculature, which is reverted when the protein is restored in a murine model of oxygen-induced retinopathy. This study associates the relevance of Ang-1 in the retinal angiogenesis during the hypoxic states in retinal development [39].

### 3.4. Astrocytic Epoxyeicosatrienoic Acids Are Proangiogenic Factors

Epoxyeicosatrienoic acids (EETs) are synthesized from arachidonic acid by the activity of cytochrome P450 epoxygenases (CYP) and participate as signaling molecules in a wide range of physiological functions to regulate vasomotor tone [90,91].

The signaling cascades activated by EETs have also been related to the angiogenic process through the regulation of ECs migration and proliferation [92]. In this context, it is important to note that astrocytes produce and release EETs that induce cerebral arteriole vasodilation [90].

In relation to angiogenesis, an increased CYP activity in cerebral areas where capillary density is higher in adult mice has been observed [93]. Moreover, in vitro studies using cocultures of cerebral microvascular ECs and astrocytes have shown that EETs released by astrocytes stimulates the proliferation and tube formation of ECs, and this was significantly diminished by CYP inhibitor [94,95], suggesting that EETs acts in a paracrine manner stimulating cerebral angiogenesis. These effects of astrocytic EETs on cerebral EC have been suggested to be related to the activation of tyrosine kinase cascades as Ras-mitogen activated protein kinase kinase-mitogen activated protein kinase (Ras-MEK-MAP), and VEGF through MAP kinase after its binding to fetal liver kinase (Flk) receptor [95].

Consistent with this, Harder et al. (2002) observed that astrocytes in coculture with ECs induce the formation of capillary-like structures, and suggested that glutamate released from neurons stimulate EETs production from arachidonic acid in astrocytes, the subsequent release of growth factors by astrocytes, and the proangiogenic effects in ECs [96]. Furthermore, in human retinal astrocytes, hypoxia induces *CYP2C* expression and augments VEGF production in astrocytes, and the inhibition of CYP reduces VEGF levels in the retina. Interestingly 11,12-EET induced the ECs’ proliferation and tube formation, but a general inhibitor of CYP did not reduce the proangiogenic effect of VEGF in ECs, suggesting that effects of EET are mediated by VEGF secretion from astrocytes [97]. Thus, it seems that EETs produced by astrocytes indirectly stimulate angiogenesis through VEGF signaling in the brain and retina.

### 3.5. Possible Role of Meteorin in Vascular Consolidation

Meteorin is a protein expressed not only by neural progenitors but also by astrocytes, and it participates in the glial cell differentiation process and in the axonal extension network during neurogenesis [98]. Additionally, Meteorin has been shown to be involved in GFAP-positive glial differentiation during embryonic brain development, and its expression is maintained in GFAP-positive glial cells even when it disappears in neurons or neural stem cells [99]. Although Meteorin may promote the differentiation of astrocytes in brain development and its expression has been described in the astrocytic endfeets that encircle the blood vessels in the postnatal mouse brain, a negative role in the angiogenic process has been reported. Meteorin reduces angiogenesis by increasing the expression and release of the antiangiogenic glycoproteins thrombospondin-1/-2 (TSP-1/-2) from astrocytes after the activation of extracellular signal-regulated kinase 1/2 (ERK1/2) and the Akt1 pathway by Meteorin [100]. Furthermore, the antiangiogenic signal mediated by astrocytes through TSP-1/-2 is consistent with other studies that showed potent inhibition of endothelial survival, proliferation, and migration, even antagonizing the effect of VEGF [101,102]. Although this seems to be a contradictory mechanism, Meteorin and its downstream mediator TSP-1/2 are important for vascular consolidation and maturation, highlighting the relevant and complex process mediated by astrocytes in angiogenesis [100].

The principal molecules secreted by astrocytes that are modulated to angiogenesis, described previously, are represented in Figure 2.

In addition to the astrocytes released by astrocytes that act in ECs promoting angiogenesis, a crosstalk between ECs and astrocytes that regulates astrocyte differentiation will be described in the next section.

## 4. Feedback between Astrocytes and Blood Vessels

As mentioned above, astrocytes stimulate the proliferation and migration of ECs, allowing the formation of new blood vessels; however, ECs in turn send signals to astrocytes, allowing their proliferation and/or maturation during angiogenesis.

### 4.1. Participation of the Apelin/APJ System in Astrocyte Network Formation in an Endothelial-Dependent Manner

Studies in retinal models have been performed to clarify the communication between ECs and astrocytes during angiogenesis. A study by Sakimoto et al. (2012) showed that apelin, a ligand for the G protein-coupled receptor APJ, which is expressed in ECs, is necessary for astrocyte maturation through a mechanism dependent on leukemia inhibitory factor (LIF) secreted from ECs [103]. This finding is consistent with the study of Kubota and Suda, where LIF showed to be an important mediator of proper vascular network formation in genetically modified mouse retina models, and it is part of a complex negative feedback mechanism with oxygen that reduces VEGF release from astrocytes [11]. These observations are supported by studies in which APJ or apelin deficiency promoted delayed angiogenesis, LIF deficiency eliminated APJ/apelin signaling and generated immature astrocytes, which hindered vascular organization, and an abnormal astrocyte network was observed during retinal formation [3,103].

Previous studies have addressed the positive role of LIF in astrocyte differentiation [104,105]. Interestingly, the activation of the Ang receptor, Tie2 in ECs, promotes the production of apelin from ECs and the maturation of blood vessels [106]. These findings suggest that the apelin/APJ system and LIF signaling, possibly mediated by Tie2 activation in ECs, are part of feedback signals between astrocytes and ECs that regulate their simultaneous maturation in order to stabilize the new blood vessels. Whether this crosstalk mechanism also operates during brain angiogenesis remains to be demonstrated.

### 4.2. Endothelial Yes-Associated Protein 1 and Astrocyte Scaffolding Formation

Yes-associated protein 1 (YAP 1 or YAP) is a transcriptional coactivator essential for endothelial behavior and retinal angiogenesis [107]. YAP/Transcriptional coactivator with PDZ binding motif (TAZ) are activated in sprouting retinal ECs by VEGF signaling and VE-Cadherin, and in turn regulates actin cytoskeletal contractility, cell adhesion and cell migration [108].

YAP was shown to be a critical regulator of angiogenic sprouting in mouse retinal vessels in a manner dependent on VE-cadherin and Ang-2, probably as a transcriptional target of YAP, and it has been suggested that disruption of VE-cadherin in ECs would activate YAP nuclear translocation and transcriptional activity [109]. In fact, the sustained binding of YAP/TAZ to the transcription factor signal transducer and activator of transcription 3 (STAT3) was shown to promote angiogenesis in a manner dependent on the increase in Ang-2 expression in ECs [110,111]. Moreover, the endothelial YAP deletion reduced the LIF secretion, caused a disruption of the astrocytic network, and reduced the astrocyte differentiation. In addition, in astrocyte cultures treated with conditioned medium from ECs that overexpressed YAP, increased astrocyte differentiation was observed [111]. This finding suggests that YAP regulates the LIF secretion from ECs that promote the maturation of astrocytic networks. Despite the mechanisms related to YAP activation and signaling remains to be elucidated. Considering the secretion of Ang-1 from astrocytes and its participation in angiogenesis, it is possible to hypothesize that Ang-1 secreted by astrocytes activates Tie-2 receptors in ECs, first promoting their migration following the astrocytic cues and then when sprouting ECs contacts astrocytes, their response with an increase of LIF production and secretion in a apelin/APJ-dependent manner and through YAP signaling induced by VE-Cadherin disruption in ECs, that finally stimulates astrocyte maturation and stabilization of the vascular plexus. If this mechanism is part of the crosstalk during retina and brain angiogenesis remains to be elucidated.

### 4.3. Oxygen Levels and Hypoxia-Inducible Factors Provide Essential Cues for Astrocyte Behavior during Angiogenesis

Oxygen levels were shown to modulate astrocyte proliferation and maturation in the brain and in the retina [112,113]. Consistent with this idea, proliferating retinal astrocytes are distributed in avascular areas and near the leading edge from the expanding vascular network where oxygen levels are low, while GFAP+ astrocytes, which indicate differentiation of these cells, are located in areas covered with blood vessels. Additionally, the high oxygen atmosphere newborn mice were raised in prevented the vascularization in the retina, enhanced astrocytes proliferation and the VEGF production, and also inhibited their differentiation. This finding suggests that blood vessels affect the proliferation and/or differentiation of astrocytes through feedback signals released that depend on oxygen levels [114]. Furthermore, it was observed that blood vessels formation slightly precedes astrocytic precursors differentiation, but no direct contact between blood vessels are needed to induce astrocytic differentiation [115].

Additionally, in HIF-2a-deficient mice, an accelerated astrocytic differentiation was observed, suggesting that oxygen diffusion from blood vessels might induce astrocyte differentiation, in part by triggering HIF-2a degradation in astrocytic progenitor cells [57]. Interestingly, deletion of HIF-1 in neuroretina reduces the organization and proliferation of the astrocytes network, an effect that depends on platelet-derived growth factor (PDGF) signaling, suggesting that neuroretina an nor astrocyte acts as oxygen sensor increasing and/or decreasing PDGF secretion and modulating angiogenic astrocyte template and angiogenesis [112]. It is possible that variations in oxygen levels modulate the astrocytic scaffold formation and stabilization through HIF-1 levels in astrocytes and neural cells.

The representation of principal components in the crosstalk signaling between astrocytes and ECs during the angiogenic process, mentioned above, is summarized in Figure 3.

The potential mechanisms described previously are associated with the angiogenesis in the brain and in retina, but it is plausible to suggest that their dysregulation could be associated with the progression of many neurological diseases, including brain cancer [116]. Cancer is one of the leading causes of death worldwide, and its hallmark is an uncontrolled cell proliferation [117]. Brain tumors are not the principal type of cancer but cause a significant morbidity and mortality in all ages, underlying the necessity of finding new targets for their treatment. The next section attempts to establish the differences in astrocytes behavior in pathological angiogenesis associated with one of the most aggressive types of brain cancer, glioblastoma.

## 5. A Possible Role of Astrocytes in Tumoral Angiogenesis and Signaling Pathways Involved in Glioblastoma

In the brain, the tumors consist of a heterogeneous group of diseases, and the different kinds of cells present in this organ, which have integrated and complex functions, making brain cancer a devastating disease. Sadly, the high grade of vascularization in the brain, according to its elevated metabolic demand, and the disruption of the blood brain barrier by tumoral cells have been associated to metastatic pattern, which reduces the patients survival prognosis [118,119,120]. Some studies have highlighted the relevance of the requirement of oxygen and nutrients to supply the tumor development through the formation of new blood vessel to provide them, and therefore during the last decades tumoral angiogenesis has been targeted to propose new therapies [121].

Glioblastoma or astrocytoma is the most common and aggressive primary brain tumor (inside the glioma tumors), and the most fatal primary brain cancer found in children and adults [122,123], and current therapeutic strategies have poor efficacy [124]. In addition, the glioblastoma environment is composed of reactive astrocytes, fibroblasts, vascular pericytes, immune cells, microglia/macrophages, and ECs. Astrocytes in contact with glioma cells in the tumoral microenvironment develop a reactive phenotype that involve the secretion of several factors that stimulate glioma cell invasion and progression [125].

In glioblastoma, it has been described that fibronectin and integrin α5β1 interaction induces retraction fibers in glioblastoma cells and promotes their dissemination and infiltration, suggesting they are involved in the establishing of a metastatic state [126]. Glioblastoma cells have direct contact with ECs, and tumor infiltration occurs in invasive niches between blood vessels and reactive astrocytes [127]. Interestingly, astrocytes with mutated forms of the tumor suppressor p53 protein had high levels of fibronectin and laminin, and glioblastoma cells also block the expression of p53 in astrocytes [128]. This suggests that the secretion of extracellular components from astrocytes in the tumoral microenvironment which promote glioblastoma migration depends on a crosstalk with tumoral cells. Furthermore, it is possible to hypothesize that fibronectin secretion from tumoral astrocytes also promotes angiogenesis, in a similar manner than occurs in physiological conditions, increasing even more the glioblastoma aggressivity.

In addition, glioma cells express N-cadherin and its expression has been related to changes in the dissemination and invasiveness of tumors as high grade as glioblastomas and patient poor prognosis [129]. A decreased levels of N-cadherin protein and subsequent activation of small GTPase Cdc42 mediated by integrin promotes glioma cells polarization and migration [130], which is consistent with a study from Asano et al. where the N-cadherin expression inversely correlated with invasion of glioma cells in a rat model [105]. Despite this seems contradictory with other studies that demonstrated an increase of mRNA levels of N-Cadherin in glioma cells [131,132]. It has been suggested that N-cadherin cleavage occurs by the cell surface metallopeptidase ADAM-10 and could explain the reduction in N-cadherin protein associated with glioblastoma dissemination and infiltration [133,134]. Moreover, the upregulation of N-Cadherin is related to the radioresistance of glioblastoma cells, by increase of the cell-cell interaction and reduced proliferation, which is consistent with the reverse effects observed when the N-cadherin expression was abolished by CRISPR/Cas9 method [135]. As described in Section 2.1, N-cadherin reduction has been associated with oligodendrocytes migration through astrocyte template [18], and it is possible to suggest that an initial disruption of N-cadherin signaling in glioma cells allow them to migrate using the astrocytic template, and then an increase of N-cadherin expression reestablish cell to cell contacts between these cells or with astrocytes, promoting resistance to the treatments. However, more studies are needed to show a definitive role of N-cadherin in glioblastoma.

Further of the classical N and R cadherins, the atypical cadherin Fat1 could be involved in glioblastomas development. The expression of the Fat1 cadherin is variable among different types of cancer and also depending on the type of tumor has demonstrated to be oncogenic or tumor suppressor [136]. In glioma cells a high expression of Fat1 dependent on the activation of NFκB signaling was observed [137]. The deletion of Fat1 signaling reduce glioma cells migration and invasiveness, an effect that was related to an increase of the expression of tumor-suppressor gene programmed cell death 4 (PDCD4) [138]. Moreover, in glioma cells in hypoxic conditions, an augmentation of the expression of Fat1 induce an increase of HIF-1 expression through growth factor receptors-Akt-mTOR pathway [139], suggesting that Fat1 controls the invasiveness of glioma cells through HIF-1 signaling. Interestingly, in recent studies it has been described that expression of Fat1 cadherin in astrocytes during postnatal angiogenesis is necessary for the association of ECs and astrocytes, proliferation and migration of astrocytes progenitors and their transition into immature astrocytes (see Section 2.1) [25]. According to this, it is possible to propose that in the tumoral hypoxic microenvironment, an increase of Fat1 signaling in astrocytes and glioma cells contributing to tumoral migration and tumoral angiogenesis through HIF-1 signaling conduiting to an aggressive tumor pattern.

As has been previously described, astrocytes release numerous molecules including VEGF, FGF-2, EETs and Ang-1 that participate in angiogenesis and could be involved in brain cancer and specifically in glioblastoma. It is known that VEGF signaling is essential for angiogenesis under physiological and pathological conditions and the inhibition of VEGF/VEGFR has been targeted as a potential treatment in various types of cancers [140,141].

In glioma cells, VEGF stimulates their proliferation and tumorigenicity through the VEGFR2 signaling pathway [142,143]. Different strategies have been used to inhibit VEGF signaling in glioblastoma, anti-VEGF antibody therapy, and tyrosine kinase receptor inhibitors, alone or in combination with other conventional therapeutic strategies, and have had dissimilar results, including prolonging survival in some patients or no benefits in terms of overall survival, and a resistance to anti-VEGF therapy has also been observed in some patients [144]. Interestingly, when a treatment with antibodies against VEGF was used, despite the reduction in tumoral angiogenesis, an increase of infiltrating tumor cells correlated with an increase of HIF expression in glioma cells was observed [145], suggesting that hypoxic environment due loss of angiogenesis and mechanisms dependent on HIF-1 signaling could account for tumor progression in invasive areas. In accordance with this, in glioblastoma patients treated with VEGFR inhibitors, an increase of FGF-2 levels was observed in correlation with tumor progression [146], suggesting that targeting FGF could be part of the treatment for glioblastomas resistant to anti-VEGF therapy [147,148].

Extracellular vesicles derived from glioblastoma cells stimulates astrocyte migration and the release of FGF, VEGF and other factors from astrocyte which in turn promotes tumor cell growth [149]. Furthermore, FGF-2 and VEGF expression in tumoral astrocytes and ECs was correlated to endothelial proliferation, tumoral angiogenesis and the degree of glioma malignancy [150,151]. Consistent with the role of FGF-2 secreted by astrocytes in angiogenesis in physiological conditions that we previously described, it is possible to propose that in the tumoral microenvironment the glioma cells induce even a greater increase in FGF-2 in astrocytes, promoting angiogenesis through FGF signaling in ECs and account for the high grade of vascularization of glioblastomas. Also, it has been described that FGF-2 secreted by glioma cells augments the function of blood barrier function and contributes to therapy resistance [152], suggesting that FGF-2 has a further role in blood vessels already formed in the perivascular niche.

Additionally, Angs has been involved in angiogenesis in glioblastomas, where the expression of Ang-1 was in the tumoral cells and also in astrocytes, and it was demonstrated that Ang-1 secreted by glioma cells promotes angiogenesis in vitro [153]. Further, in glioblastomas, a correlation of the cell proliferation and an overexpression of Astrocyte elevated gene (AEG)-1 has been observed [154]. This is interesting to note because AEG-1 also induce tumoral angiogenesis through an increase of the proangiogenic factors Ang-1, HIF-1 and matrix metalloproteinases via PI3K/Akt signaling, and the blockade of Tie-2 receptors inhibits ECs angiogenesis in vitro [155]. Considering that Ang-1 secreted from astrocytes stimulates physiological angiogenesis under hypoxic conditions (see Section 3.3), it is possible to suggest that Ang-1 released by astrocytes and glioma cells acts in a coordinated manner activating Tie-2 signaling in ECs to promote the highly vascularized phenotype of glioblastoma, and at least part of the signaling depends on AEG-1 and HIF-1, permitting the transition since a glioma tumor (less aggressive) to glioblastoma (highly aggressive).

In summary, these antecedents support the notion that glioblastoma development has to be considered in the context of multiple interactions of signal molecules between ECs, astrocytes, and glioma cells that modulate tumor progression and angiogenesis. Some of the astrocytic mediators, which participate in the brain postnatal angiogenesis under physiological conditions, seem to be altered in glioblastoma, usually relating their overexpression to the progression, severity, and aggressiveness of this type of tumor. These observations make them interesting as possible targets and could establish a multifunctional therapeutic treatment to the poor prognosis of the glioblastomas diagnosis.

The interactions between glioblastoma, astrocytes, and ECs that promote angiogenesis and tumor progression, described above, are represented in Figure 4.

## 6. Conclusions and Perspectives

There seems to be a consensus on the importance of the role of the astrocytes during angiogenesis in the brain development and in the retina. Astrocytes assemble as scaffolding and release factors that modulate ECs migration promoting angiogenesis. Moreover, a crosstalk signaling between ECs and astrocytes induce astrocyte differentiation when newly blood vessels are completely formed, and oxygen levels start to rise. Despite great efforts to establish the interaction between all the mediator related in the signaling associated to the angiogenesis described above, future studies are required to clarify the interrelation between the signaling pathways involved in the endothelial cell-astrocyte interaction, as well as the possible communication with other cell types that participate in angiogenesis.

The participation of neurons during angiogenesis has been addressed in numerous studies and escapes the description given in this review; it is possible that signaling from neurons regarding the stimulation of the release of astrocytic proangiogenic mediators and the establishment of the astrocytic scaffold during angiogenesis occurs in parallel and maybe the signaling of ECs, astrocytes, and neurons converge in a common complex of sequential steps required for the adequate consolidation of the vasculature in the CNS and in the retina. New studies will be required in order to clarify the crosstalk signaling between these three types of cells during the angiogenic process.

Although it is not surprising, some of the pathways described in astrocytes during physiological angiogenesis are altered during the development of one of the main and most aggressive brain tumors, glioblastoma. We believe that certain components described in this review and mainly associated with changes in the communication between astrocyte, ECs, and glioma cells could represent future targets for the treatment of this devastating disease. Future studies are required to better elucidate the mechanisms of interrelationship between these three types of cells, and of course other cell types that constitute the microenvironment of the glioblastoma.

## Figures and Tables

**Figure 1 ijms-23-02646-f001:**
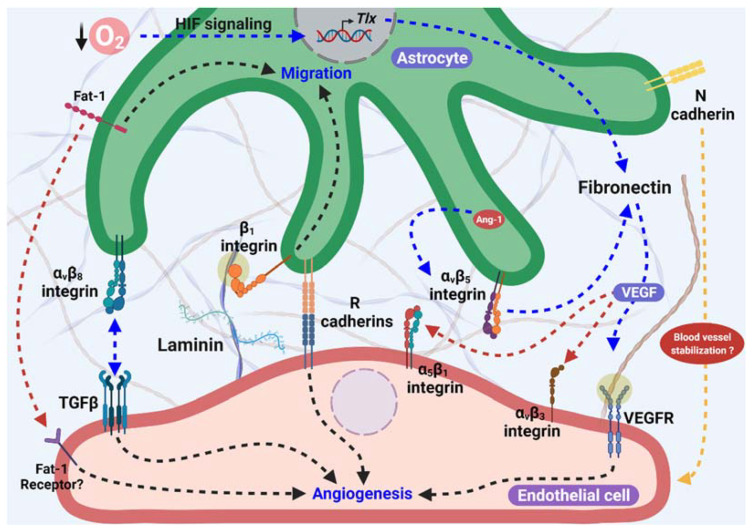
Astrocytes as proangiogenic scaffolds. R-cadherin in the astrocyte process surrounding the endothelial filopodia extension forms small clusters, and promotes angiogenesis. In addition, Fat-1 cadherin from astrocytes is necessary for endothelial cells (ECs)-astrocytes interaction, and also to proliferation and maturation of astrocyte progenitor cells in the astrocyte template. Furthermore, Fibronectin production in astrocytes is stimulated by lower O2 levels in avascular areas through HIF and TLX signaling and/or in an Ang-1 dependent manner, which activate α5β1 and αvβ3 integrins and VEGFR2 signaling, inducing endothelial migration. αvβ8 integrin from astrocytes is necessary for ECs migration dependent on the activation of TGFβ signaling. Additionally, laminin-β1 integrin induces astrocyte migration and promotes astrocyte scaffolding formation. Moreover, the participation of N-cadherin in blood vessel stabilization has been suggested.

**Figure 2 ijms-23-02646-f002:**
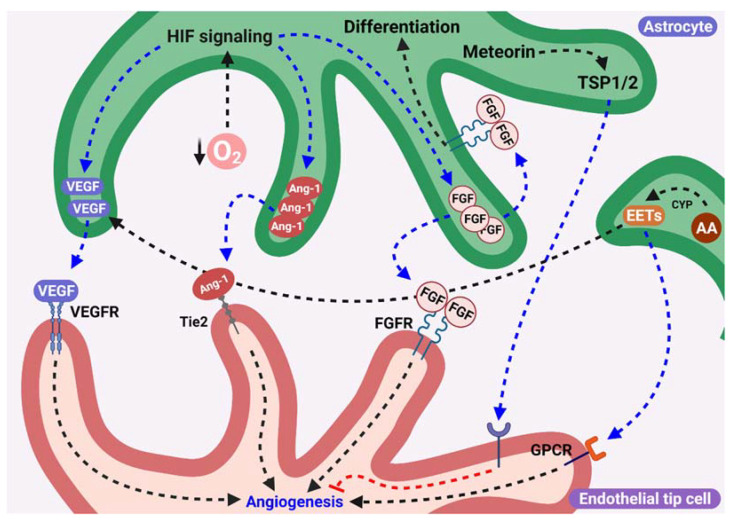
Proangiogenic factor released by astrocytes. Astrocytes that surround endothelial tip cells synthesize and secrete proangiogenic factors such as VEGF, FGF, Ang-1, and EETs. Hypoxia stimulates the secretion of VEGF and FGF-2 from astrocytes that surround endothelial tip cells, which in turn activate their receptors (VEGFR2 and FGF2R, respectively) in ECs inducing angiogenesis. Additionally, FGF acts as an autocrine factor inducing astrocytes differentiation. The astrocyte secretion of Ang-1 induced by hypoxia stimulates angiogenesis through Tie2 activation in ECs. Moreover, EETs production in astrocytes in a CYP dependent manner stimulates angiogenesis through VEGF secretion from astrocytes or through activation of G-protein coupled receptor (GPCR) in ECs. However, astrocytes surrounding ECs from blood vessels are stabilized after their formation by secreted Meteorin, which induces the secretion of thrombospondin 1/2 (TSP1/2), and inhibits angiogenesis inducing the consolidation of newly formed blood vessels.

**Figure 3 ijms-23-02646-f003:**
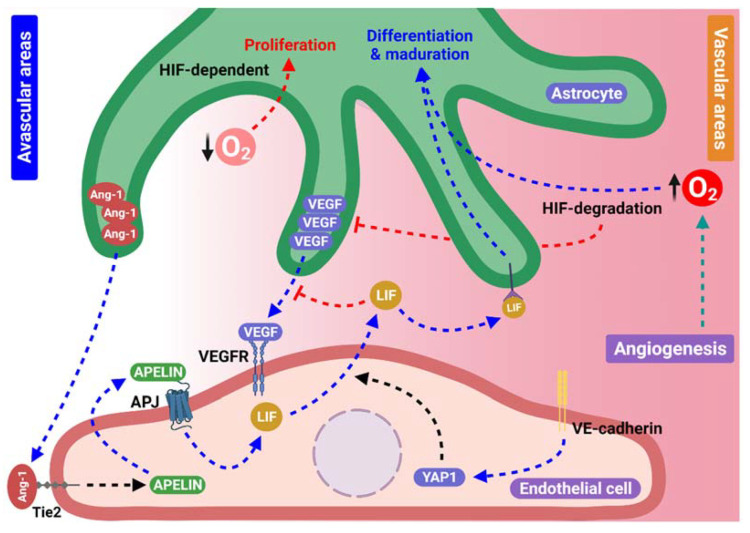
Feedback between astrocytes and endothelial cells. Production and secretion of Apelin dependent on Ang/Tie2 signaling in ECs activates the APJ receptor, and induces leukemia inhibitory factor (LIF) secretion. In addition, LIF secretion can also be induced by YAP1 possibly dependent on VE-cadherin or Fat1 signaling and stimulate the differentiation and maturation of astrocytes that surround newly formed blood vessels, reducing VEGF secretion. Furthermore, oxygen levels can induce differential processes in astrocytes, while in avascular areas, a low oxygen concentration induces astrocyte proliferation to form angiogenic scaffolds in a HIF-dependent manner. In areas near new blood vessels, oxygen concentrations start to rise and induce differentiation and maturation probably through HIF degradation.

**Figure 4 ijms-23-02646-f004:**
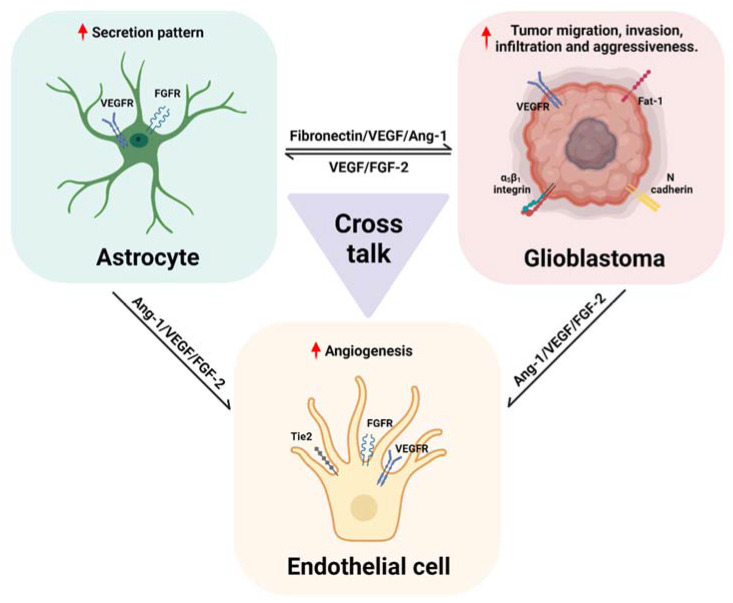
Reciprocal signaling between astrocytes, endothelial cells and glioblastoma cells promotes tumor progression and angiogenesis. Fibronectin secreted from astrocytes mediated by inhibition of p53 expression induced by glioblastoma cells, stimulates tumor dissemination through α5β1 integrin interaction in glioblastoma. Moreover, a reduction in N-Cadherin mediated by ADAM-10 in glioblastoma cells stimulates their migration. An increased expression of the atypical cadherin Fat1 promotes the invasiveness of glioblastoma cells, through an increase of HIF-1 signaling and perhaps stimulates angiogenesis through Fat1 signaling in astrocytes as was observed in physiological conditions. Similar to what was observed in astrocytes, glioblastoma cells in hypoxic conditions secrete VEGF, presumably dependent on HIF-1 signaling, that acts in an autocrine mode, stimulating tumor progression and promoting angiogenesis via VEGFR signaling in ECs. Furthermore, VEGF and FGF secretion from astrocytes is stimulated by extracellular vesicles derived from glioblastoma, and could also stimulate angiogenesis and tumor cell growth, highlighting the intricate signaling that leads to an increase in these proangiogenic factors and their signaling in the tumor microenvironment. Comparable to that described in astrocytes during physiological angiogenesis, Ang-1 is secreted by glioblastoma cells in hypoxic conditions, an effect mediated by Astrocyte elevated gene (AEG)-1 signaling, which in turn increases HIF-1 and Ang-1 levels, and promotes angiogenesis mediated by Tie-2 signaling.

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
