# Peer review of "Key Role of Astrocytes in Postnatal Brain and Retinal Angiogenesis"

_ijms, 2022, doi:10.3390/ijms23052646_

Round 1
Reviewer 1 Report
This publication complies with the profile of Special Issue Molecular Basis of Vascular Remodeling 2.0 and may be published after a number of important corrections have been made.
- First of all, the title and abstract of the article do not fully correspond to its content. This review is based on data on the mouse retinal model and retinal angiogenesis. Therefore, this should be reflected in the title of the publication and in the abstract.
- If the authors limit themselves to considering the processes associated with retinal angiogenesis, then they should be described in more detail. The review lacks a number of recent data, such as the involvement of Fat1 in this process.
- It is necessary not only to enumerate possible molecular pathways, but also to analyze their contribution to this process and evaluate their possible involvement in pathological processes (eg, cancer) or regeneration processes.
- It is also necessary to compare the processes that occur during the interaction of astrocytes and endothelial cells in the retina and the brain.
- Many sections devoted to the description of individual signaling pathways and intercellular interactions are not described in sufficient detail, for example, Fibronectin and Astrocytic epoxyeicosatrienoic acids.
- When describing astrocytes as oxygen sensors, one should not be limited only to the description of TLX.
- The contribution of the spectrum of proangiogenic markers to the development of retinal angiogenesis is not fully shown. In particular, there are no data on the repertoire of FGFR receptors.
Author Response
Dear reviewer 1,
First of all, we want to thank you for your comments on our review, which have helped us to better address the interrelationship between astrocytes and endothelial cells during angiogenesis and extend our review to a pathological perspective. We added some modifications in the text and in the figures of the article using the tracking changes tool in order to address your comments.
Best regards,

Reviewer 2 Report
In this paper, the authors talked about the relationship between astrocytes and endothelial cells. With diagrams, the authors clearly illustrated the role of relating molecules, like VEGF, FGF, Ang-1, to the migration and proliferation of endothelial cells and the happening of the angiogenic process.
Well in writing and organizing, clear in method, Authors covered most of the factors in the angiogenic process.
Author Response
Dear Reviewer 2,
First of all, we want to thank you for your comments on our review. We added some modifications in the text and in the figures of the article using the tracking changes tool in order to address the comments from reviewer 1.
Best regards,
Round 2
Reviewer 1 Report
All comments have been corrected.